# Lectin-Based Affinity Enrichment and Characterization of N-Glycoproteins from Human Tear Film by Mass Spectrometry

**DOI:** 10.3390/molecules28020648

**Published:** 2023-01-08

**Authors:** Carsten Schmelter, Alina Brueck, Natarajan Perumal, Sichang Qu, Norbert Pfeiffer, Franz H. Grus

**Affiliations:** Department of Experimental and Translational Ophthalmology, University Medical Center, Johannes Gutenberg University, 55131 Mainz, Germany

**Keywords:** N-glycoproteins, tear film, multi-lectin column, PNGase F, mass spectrometry

## Abstract

The glycosylation of proteins is one of the most common post-translational modifications (PTMs) and plays important regulatory functions in diverse biological processes such as protein stability or cell signaling. Accordingly, glycoproteins are also a consistent part of the human tear film proteome, maintaining the proper function of the ocular surface and forming the first defense barrier of the ocular immune system. Irregularities in the glycoproteomic composition of tear film might promote the development of chronic eye diseases, indicating glycoproteins as a valuable source for biomarker discovery or drug target identification. Therefore, the present study aimed to develop a lectin-based affinity method for the enrichment and concentration of tear glycoproteins/glycopeptides and to characterize their specific N-glycosylation sites by high-resolution mass spectrometry (MS). For method development and evaluation, we first accumulated native glycoproteins from human tear sample pools and assessed the enrichment efficiency of different lectin column systems by 1D gel electrophoresis and specific protein stainings (Coomassie and glycoproteins). The best-performing multi-lectin column system (comprising the four lectins ConA, JAC, WGA, and UEA I, termed 4L) was applied to glycopeptide enrichment from human tear sample digests, followed by MS-based detection and localization of their specific N-glycosylation sites. As the main result, our study identified a total of 26 N glycosylation sites of 11 N-glycoproteins in the tear sample pools of healthy individuals (*n* = 3 biological sample pools). Amongst others, we identified tear film proteins lactotransferrin (N497 and N642, LTF), Ig heavy chain constant α-1 (N144 and 340, IGHA1), prolactin-inducible protein (N105, PIP), and extracellular lacritin (N105, LACRT) as highly reliable and significant N glycoproteins, already associated with the pathogenesis of various chronic eye diseases such as dry eye syndrome (DES). In conclusion, the results of the present study will serve as an important tear film N-glycoprotein catalog for future studies focusing on human tear film and ocular surface-related inflammatory diseases.

## 1. Introduction

Human tear film is a complex mixture of various proteins, electrolytes, metabolites, and lipids covering as well as wetting the ocular surface, cornea, and conjunctiva [1]. It has a thickness of 3 µm, comprises a volume of 7–10 µL, and performs important functions such as the delivery of oxygen and nutrients to the cornea or the immune protection of the ocular surface [1,2]. The proper formation of tear film mirrors the health condition of the ocular surface, whereas any abnormality in composition or function might result in inflammation and the development of chronic eye diseases such as dry eye syndrome (DES) [3]. Considering these facts, tears are a valuable source for biomarker discovery since they are easy to access and can be taken by non-invasive sample collection methods. Thereby, basal tears are either collected by capillary tubes or by Schirmer’s strip technology, offering both certain advantages as well as disadvantages in routine clinical practice (as summarized in [4]). However, tears as a potential biomarker pool are not restricted to pathologies affecting the ocular surface (e.g., DES [5,6]) or other eye-related structures (e.g., the retina, as in glaucoma [7]), but are also suitable for biomarker discovery in other neurodegenerative or systemic diseases including Alzheimer’s disease (AD) [8], Parkinson’s disease (PD) [9], multiple sclerosis [10], or even cancer [11].

Protein glycosylation is one of the most common post-translational modifications (PTMs), playing a fundamental role in many biological processes such as protein function, stability and degradation, cell signaling, immune responses, and cell cycle regulation [12,13]. This specific PTM also has important functions in maintaining ocular surface homeostasis and forming a natural protective barrier against various pathogens and other microorganisms [14]. Accordingly, pathogenic glycomic changes have already been documented in the basal tears of patients with diabetic retinopathy [15] as well as keratoconjunctivitis [16], demonstrating the immense power of protein glycosylations as a sensitive indicator for biomarker discovery and early diagnosis. In general, protein glycosylations can be divided into O-linked and N-linked glycosylations depending on the side chain of the amino acid (aa) at the respective PTM site [17]. Therefore, N-glycosylations naturally occur at asparagine (N) residues exhibiting the canonical protein sequence motif N–X–serine (S)/threonine (T)/cysteine (C), where X indicates any aa except proline (P) [17]. In contrast, O-glycosylations can occupy any S as well as T residue without any obligatory canonical protein sequence [17]. Furthermore, N-glycosylations can be enzymatically released from the protein backbone by peptide N-glycosidase F (termed PNGase F), facilitating the independent analysis of N-glycan motifs and the corresponding protein N-glycosylation site [18]. In this context, the enzymatic release of the N-glycan results in the deamidation of N to D (aspartic acid), favoring the detection of the original N-glycosylation site with a mass shift of +0.984 Da as analyzed by mass spectrometry (MS) [19]. However, the enzymatic deglycosylation of O-glycoproteins is not feasible and can be only achieved by chemical reactions such as β-elimination [20]. The disadvantage of this chemical deglycosylation process is the occurrence of peeling reactions, resulting in undesirable modifications of the natural O-glycan motifs and the respective protein sequence [20,21]. Because of this, the present study focused on the detection and characterization of N-glycosylation sites in human tear film which will serve as the basis for ocular biomarker discovery in the future. Moreover, previous studies examining the glycomic composition of basal tears mainly detected about 50–150 N-glycans compared to 8 O-glycan motifs [15,16], confirming the predominant presence and regulatory functions of this glycosylation type in human tear film.

The main objective of the present study was to develop a lectin-based affinity purification method for the enrichment of glycoproteins/glycopeptides from human tear film and the MS-based detection as well as localization of their specific N-glycosylation sites. First, we established and optimized the lectin-based enrichment technology on the native tear glycoprotein level by 1D gel electrophoresis and specific protein stainings. Afterward, we applied the best-performing lectin column system for tear glycopeptide enrichment from healthy individuals, followed by characterization of their specific N-glycosylation sites via high-resolution MS (see the schematic workflow in Figure 1). To the best of our knowledge, this is the first study investigating protein N-glycosylation profiles in tear samples derived by the Schirmer’s strip collection method. The results of the present project will serve as an important N-glycoprotein reference map for future studies focusing on the ocular surface and will help to unravel the complex regulatory functions of N-glycoproteins in maintaining tear film stability and the ocular defense system.

## 2. Results

Within the scope of the present study, we developed an affinity method to enrich glycoproteins, or rather glycopeptides, from human tear sample pools of healthy individuals. For enrichment of the tear glycoproteins and glycopeptides, we used the principle of lectin affinity chromatography (see the workflow in Figure 1) which is based on the specificity of lectins for particular glycan motifs (e.g., α-linked mannose). For this purpose, we first evaluated the performance of various single-lectin as well as multi-lectin columns regarding the efficiency to enrich native glycoproteins from human tear sample pools. This was assessed by 1D gel electrophoresis of the enriched (glyco-)proteins and the execution of specific protein stainings (see method Section 4.3 and Section 4.4). After the successful affinity method development and optimization, we applied this established technology for the enrichment of glycopeptides from human tear sample digests (see method Section 4.5). In the end, we characterized the enriched tear glycopeptides and their specific N-glycosylation sites by high-resolution MS.

### 2.1. Enrichment of Tear Glycoproteins and Affinity Method Development

First, we evaluated the enrichment performance of single-lectin columns (ConA, JAC, and WGA) as well as a multi-lectin column device (combination of the lectins ConA, JAC, and WGA, termed 3L) on the native glycoprotein level. The lectin ConA showed a high affinity for α-linked mannose, JAC for galactose ß(1–3)-linked to N-acetylgalactosamine, and WGA for N-acetylglucosamine. The enriched glycoproteins (eluate fractions) of the different lectin column systems were separated by 1D gel electrophoresis and subsequently stained with the Coomassie protein dye (see Figure 2). Furthermore, we included an untreated tear sample pool (50 µg) in this analysis to match potential enriched glycoprotein spots with the native tear film proteome. In addition, we marked the most abundant protein markers (e.g., Ig α-1 C chain at ≈49 kDa), which were revealed by MS analysis and already intensively described in previous studies of our group [22,23], to the respective mass range.

Based on the specific protein migration pattern of the respective eluate fractions, it was observed that each single-lectin column (ConA, JAC, and WGA) as well as the 3L multi-lectin column (combination of ConA, JAC, and WGA) successfully enriched various potential glycoproteins from the human tear sample pools. Each single-lectin column effectively enriched the highly abundant tear film proteins lactotransferrin (LTF) and Ig α-1 C chain (IGHA1). These proteins were particularly enriched by ConA, in whose eluate fraction both protein spots were more intensively stained compared to the eluates of JAC and WGA. On the contrary, other highly abundant tear film markers, such as albumin (ALB), lysozyme C (LYZ), as well as cystatin-S (CST4), were less or not at all enriched by all three single-lectin columns (ConA, JAC, and WGA). Nevertheless, zinc-α2-glycoprotein (AZGP1) was specifically observable in the eluate fraction of ConA (above ≈38 kDa) but is also highlighted as a clear protein band in the eluate fraction of the 3L multi-lectin column system.

The other tear film proteins, lipocalin-1 (LCN1) and prolactin (PRL), were preferentially accumulated by WGA, whereas the protein spot intensities were much lesser in the eluate fractions of ConA and JAC. Finally, the proteins lacritin (LACRT) and prolin-rich protein 4 (PRP4) were increasingly enriched by all three single-lectin column systems (ConA, JAC, and WGA), but specifically, ConA led to an additional accumulation of a protein spot around ≈28 kDa. In summary, the eluate fraction of the 3L multi-lectin column system (combination of ConA, JAC, and WGA) showed a congruent protein migration pattern compared to all three single-lectin columns and seems to facilitate a specific as well as effective enrichment of many tear film glycoproteins. In addition, the multi-lectin column system approves the glycoprotein enrichment in a single working step and possesses a much faster and more efficient performance than the single-lectin column devices.

In the next part of the analysis, we were interested in if the lectin-specific enriched proteins are true glycoproteins or just represent an undesirable contamination within the respective eluate fractions (see Figure 3). We then repeated the enrichment with the 3L multi-lectin column system (combination of ConA, JAC, and WGA) and subjected the eluate fraction to further 1D gel electrophoresis and glycoprotein staining (see Figure 3A). At the same time, we also included an extended multi-lectin column device in this analysis, which comprised, in total, the four lectins ConA, JAC, WGA, and UEA I in equal shares (termed 4L). The lectin UEA I offered a high affinity for α-linked fucose, which was supposed to significantly improve the enrichment performance of the previously employed 3L multi-lectin column system. Additionally, we also treated two tear sample pools prior to lectin affinity enrichment (with 3L and 4L multi-column system) with endoglycosidase PNGase F to release the N-glycans from the tear glycoproteins.

By means of the specific glycoprotein staining of the 1D gel (see Figure 3A), it is clearly observable that the 3L as well as 4L multi-lectin column system successfully enriched real glycoproteins (magenta-stained protein spots) from the human tear sample pools. This was also confirmed by the glycosylated protein standard (horseradish peroxidase, termed +CTRL), which was detected as an obvious magenta-stained protein band at approximately 38–49 kDa and illustrates the specificity of the commercial glycoprotein staining kit. Specifically, the 4L multi-lectin column system accumulated two additional glycoprotein bands (magenta-stained) around 28 kDa (identified as LACRT), which were not present in the eluate fraction of the 3L column system. Moreover, the pre-treatment of the tear sample pools with PNGase F hampered the glycoprotein enrichment through the 3L as well as the 4L multi-lectin column systems (see Figure 3A and Appendix A). These findings indicate that the lectin affinity enrichment of the glycoproteins seems to predominantly occur through binding to their specific N-glycan motifs. This is also in accordance with the observation that no glycoprotein was detected in the flow-through (FT) fractions of the 3L- as well as 4L-column systems, indicating an excellent enrichment performance of the lectin affinity chromatography. For the final evaluation, we stained the 1D gel with Coomassie dye to visualize all proteins and to identify potential undesirable contaminations (see Figure 3B). The eluate fractions of both PNGase F-treated tear sample pools (3L + PNGase F and 4L + PNGase F) contained a slight blue protein spot around 14–17 kDa (identified as LCN1) which was not previously stained by the glycoprotein dye. Accordingly, this protein spot seems to represent an unspecific protein binding to the agarose lectin beads (to the 3L as well as the 4L column system). However, this slight protein cross-contamination was tolerated in the further methodical procedure.

Based on these results, the 4L multi-lectin column system (combination of ConA, JAC, WGA, and UEA I) led to an accumulation of a broader range of tear film glycoproteins compared to the 3L column device and was employed as an established and optimized affinity technology for glycopeptide enrichment.

### 2.2. Enrichment of Tear Glycopeptides and MS Analysis

After the successful development and evaluation of the lectin-based affinity enrichment protocol on the native glycoprotein level, we applied optimized technology for the accumulation of tear glycopeptides followed by mass spectrometric (MS) detection of their specific N-glycosylation sites. For this purpose, we first digested the respective tear sample pools by in-solution trypsin digestion (see method Section 4.5) and subjected them to further glycopeptide enrichment via the 4L multi-lectin column system (combination of ConA, JAC, WGA, and UEA I). In this part of the analysis, we investigated the glycoprotein/glycopeptide composition in the biological tear sample pools (*n* = 3) of healthy individuals to report the occurrence of specific N-glycosylation sites in the human tear film proteome during physiological conditions. However, the lectin-specific enrichment on the glycopeptide level has the advantage to neglect non-modified parts of the glycoproteins during the purification process and to analyze predominantly concentrated glycopeptides by MS. For the MS-based detection of N-glycosylation sites, the enriched glycopeptides were pre-treated with glycosidase PNGase F to release the N-glycans from the peptide backbone. Consequently, the enzymatic-induced release of the N-glycans leads to the deamidation of asparagine (N) to aspartic acid (D), which marks the original N-glycosylation site in the respective peptide backbone. This enzymatic-induced PTM site results in a mass shift of +0.984 Da and can be detected by high-resolution MS analysis. Nevertheless, deamidations of N can also occur as natural PTM sites or might be chemically induced during the sample preparation process. Thus, we also considered an additional control group (CTRL) in our experimental design which comprised enriched glycopeptides without PNGase F pre-treatment from a master tear sample pool. By matching the enzymatic-induced deamidation sites of N with the untreated CTRL group, we only identified the true and reliable N-glycosylation sites of each tear glycoprotein. Furthermore, we only accepted N-glycosylation sites (deamidations) which were present in the canonical protein sequence motif N–X–S/T/C and showed an Ascore ≥ 20 (see method Section 4.5 for further explanation). Applying these strict filtering criteria, we identified a total of 26 N-glycosylation sites of 11 N-glycoproteins in the human tear film proteome during physiological conditions (summarized in Table 1 and Appendix A). Thereby, Table 1 summarizes all tear glycoproteins with their most significant N-glycosylation sites and Appendix A contains the peptide sequence information of all identified N-glycopeptides in each biological replicate. All protein N-glycosylation sites were already reported in the literature and confirm the reliability of our obtained study results (see Table 1).

Hereinafter, Figure 4 illustrates the exemplary identification of the glycosylation sites N497 and N642 of the tear glycoprotein lactotransferrin (LTF) and the untreated CTRL group. The black arrows indicate both N-glycosylation sites (deamidations) N497 and N642 in the protein sequence LTF (see Figure 4A). In contrast, these PTM sites were not identified in the untreated, enriched glycopeptide group (CTRL), which proves the specific induction of this PTM site by glycosidase PNGase F. The red-marked amino acids in the glycopeptide sequence (see Figure 4B) indicate the enzymatic-induced deamidation of N to D, resulting in a mass shift of +0.984 Da which was detectable by MS analysis. Further exemplary N-glycosylation site identifications of zinc-α2-glycoprotein (AZGP1) are shown in Appendix A.

## 3. Discussion

The main objective of the present study was to develop an affinity purification method for the efficient enrichment of glycoproteins/glycopeptides from human tear film sample pools and to use mass spectrometric (MS)-based detection and localization to determine their specific N-glycosylation sites. For this purpose, we used lectin affinity chromatography, which is based on the specificity of the lectins for specific sugar residues [35]. For method development and evaluation, we first enriched the native glycoproteins from human tear sample pools by various lectin-affinity columns and assessed their performance as well as convenient handling by 1D gel electrophoresis and specific protein stainings. After this, we applied the best-performing lectin column system to enrich glycopeptides from the tryptic tear sample digests, followed by the MS-based detection of their characteristic N-glycosylation sites.

For the accumulation on the glycoprotein level, we first applied various single-lectin (ConA, JAC, or WGA) as well as multi-lectin column systems (combination of ConA, JAC, and WGA, termed 3L) and examined the composition of their specific eluate fractions (enriched glycoproteins) by Coomassie protein gel staining (see Figure 1). In conclusion, the 3L multi-lectin column system enabled an efficient enrichment of a broad range of glycoproteins and is clearly a superior method compared to the single-lectin column devices. It provided the accumulation of congruent glycoprotein species in a single working step and allowed for a much faster sample preparation in contrast to single-lectin column devices. Single-lectin column WGA, for instance, specifically enriched the protein band around 14–17 kDa (identified as prolactin, PIP), whereas the intensities of this protein band were clearly diminished in the eluate fractions of ConA and JAC. WGA has a high affinity for N-acetylglucosamine and might allow the conclusion that PIP is predominantly linked to glycan motifs containing this specific monosaccharide. To support this assumption, Wiegandt et al. (2018) [34] have already proven that PIP carries many highly fucosylated bi-, tri-, and tetra-antennary N-glycans in saliva as well as the seminal plasma of healthy individuals, which might explain the relatively high amounts of N-acetylglucosamine in their basic glycan structures. On the other hand, the single-lectin column device ConA particularly accumulated the protein zinc-α2-glycoprotein (AZGP1, see Figure 1 above ≈ 38 kDa), illustrating the lectin-specific enrichment of diverse tear glycoprotein subsets. Based on that knowledge, multi-lectin column devices show excellent abilities to capture a wide range of various glycoproteins/glycopeptides from human tear film and maximize the N-glycosylation site identifications in the posterior MS analyses. This is also in accordance with previous studies which successfully used different multi-lectin column systems for the successful and reproducible glycoprotein enrichment of samples from human serum [36,37], plasma [38,39,40], and brain tissues [41] as well as from other biological sources [42].

In the next part of the analysis, we were interested if the enriched protein species are real glycoproteins and if the efficiency of the current 3L multi-lectin column system might be increased by the inclusion of the further lectin species UEA I (4L multi-lectin column; a combination of ConA, JAC, WGA, and UEA I). The lectin UEA I showed a high affinity for α-linked fucose and is supposed to significantly enhance the enrichment performance of the current 3L column system. The specific glycoprotein staining (see Figure 4A) proved that both multi-lectin column systems (3L and 4L) led to the accumulation of various glycoprotein subsets from the human tear sample pools and that the affinity purification was clearly based on the binding of the lectins to the respective N-glycan structures. Furthermore, the extended 4L multi-lectin column system specifically enriched two additional glycoprotein bands around 28 kDa (identified as lacritin, LACRT) compared to the 3L column system, indicating the accumulation of a broader range of glycoproteins from the human tear sample pools. Because of this, the 4L column system was used as an optimized lectin affinity method to enrich glycopeptides from the human tear sample digests. Nevertheless, both multi-lectin column devices (3L and 4L) also accumulated proteins that were identified as non-glycosylated markers (e.g., 14–17 kDa in Figure 4B; identified as lipocalin-1, LCN1) and subsequently represent undesirable contaminants in the eluate fractions. In accordance, Yang et al. (2004) [36] also reported the occurrence of non-glycosylated albumin as slight contamination during the lectin-based glycoprotein enrichment of human serum samples. Thereby, it is assumed that the respective contaminant unspecifically interacts with the solid phase of the resin beads or represents a direct interaction partner of an enriched glycoprotein (co-elution). It has already been known for a long time that the LCN1 contaminant directly interacts with lactotransferrin (LTF) [43], which was identified as a reliable glycoprotein in the present study, and might promote this unspecific cross-contamination. However, as only low contamination was detectable in each eluate fraction (glycoprotein fraction) and further enrichment was performed on the glycopeptide level, we tolerated this finding in the following methodological procedure.

Employing the optimized lectin affinity technology (4L multi-lectin column) on the glycopeptide level, we were finally able to identify and localize 26 N-glycosylation sites of 11 tear film glycoproteins by high-resolution MS (see Table 1 and Figure 4). Each glycoprotein (or N-glycosylation site) had to be present in at least one biological tear sample pool that was originally supplied from healthy individuals. Moreover, all identified N-glycosylation sites were already described in previous publications investigating various biological fluids, such as human saliva [28,44] or human milk [24], and clarifies the reliability and accuracy of our obtained study results. To the best of our best knowledge, there was only one study so far examining the presence of N-glycosylation sites in human tear film samples [25]. Thus, the authors Zhou et al. (2009) [25] identified a total of 43 unique tear glycoproteins by MS using hydrazine resin beads for tear glycoprotein/glycopeptide enrichment. Indeed, the glycoprotein identification rate of our study seems to be relatively low in comparison to Zhou et al. (2009) [25], but several aspects should be considered in the tear sample collection method as well as the data interpretation between both study designs. First of all, the authors Zhou et al. (2009) [25] used capillary tubes for tear sample collection and glycoprotein enrichment and not Schirmer’s strips as described in our present study design. It is well known that the sample collection method also has a great impact on the protein composition as well as quantity [45] of the tear film proteome and might also influence the outcomes of the respective study. Capillary tears, for instance, contain much more extracellular proteins, compared to Schirmer’s strip-derived tear samples [45], which were identified by Zhou et al. (2009) [25] as potential N-glycoproteins (e.g., hemopexin or clusterin). In particular, N-glycoproteins are often found in extracellular environments and body fluids [46,47], which might explain the higher N-glycoprotein identification rates in the previous study. Secondly, the authors Zhou et al. (2009) [25] used the human IPI protein database for the analysis of the MS data which is currently obsolete and was replaced in 2011 by the UniProt Knowledgebase (UniProtKB) with manually reviewed protein entries in the UniProtKB/Swiss-Prot database [48] (as used in our study). This could have led to false-positive tear glycoprotein identifications due to insufficiently reviewed protein reference databases. Furthermore, we included an untreated CTRL group in our experimental study design to exclude natural or chemically-induced deamidation sites (potential N-glycosylation sites) which is also a common pitfall in large-scale N-glycoproteomic studies [49].

However, in the following sections, we will expand on the molecular function as well as bioactivity of selected tear N-glycoproteins (see Table 1) and critically discuss their potential role in the pathogenesis of diverse eye diseases, with a special focus on dry eye syndrome (DES). 

Lactotransferrin (LTF) is a highly abundant protein of human tear film [5,50] and was identified with two N-glycosylation sites (N497 and N642, see Figure 4) in the present study. Both N-glycosylation sites of LTF are very well documented in the literature and are reported to occur in several human biological fluids as well as tissues, such as capillary tears [25], saliva [44], milk [24], and the liver [26]. In general, LTF is an iron-binding glycoprotein and triggers many anti-inflammatory, immunomodulatory, and antimicrobial activities by different modes of action [51]. Thereby, these protective effects are predominantly elicited by the direct interaction of LTF with pathogenic microorganisms and are particularly mediated through its linked N-glycan motifs [52]. In accordance, Kautto et al. (2016) [53] proved that the N-glycan motifs of LTF are required for the direct interaction with *P. aeruginosa* to inhibit its ability to invade corneal epithelial cells. With respect to the eye, several studies have already demonstrated that the LTF expression levels were significantly decreased in the tear samples of DES patients compared to healthy controls [5,54,55] and might also serve as potential prognostic markers in the future [56]. Furthermore, the oral administration of recombinant lactotransferrin attenuated the inflammation of the cornea/lacrimal gland in an age-induced DES animal model and also reduced oxidative stress damage responses on the ocular surface [57]. All these findings demonstrate the important biological function of LTF to maintain the protective ocular barrier and the potential role of aberrant LTF N-glycosylation profiles in the pathophysiology of DES.

The extracellular protein lacritin (LACRT) was also identified as a reliable glycoprotein in the present study, comprising one N-glycosylation site N119 located at the C-terminus (total length 137aa). Nevertheless, this N-glycosylation site (deamidation) was also detectable in tear sample digests without previous PNGase F treatment (CTRL group), indicating a chemically-induced deamidation site occurring during the sample preparation process. However, since the N-glycosylation site N119 of LACRT was already confirmed in previous publications [25,28], we decided to accept this predicted N-glycoprotein identification. The major functions of LACRT are the stimulation of basal tear secretion, proliferation and survival of epithelial cells, and corneal wound healing (summarized in [58]). Thus, it is supposed that LACRT simulates tear production by promoting the enhanced expression of specific mucins or by activating sensory neurons [59]; however, the expression levels of LACRT were also found to be significantly down-regulated in the tears of DES patients compared to healthy controls [5,54,60]. Furthermore, the N-glycosylation levels of LACRT at site N119 were also significantly diminished in the tears of climatic droplet keratopathy (CDK) patients, indicating a general functional role of this specific N-glycosylation site in various eye disorders [25]. With respect to DES, it was already proven that the topical administration of recombinant LACRT significantly improved the basal tear production as well as clinical symptoms in different DES animal model systems [60,61] and is currently under approval as a commercial eye drop product (Lacripep^TM^ from TearSolutions Inc., Charlottesville, VA, USA) in clinical phase II [62,63]. Interestingly, synthetic Lacripep^TM^ encodes only the C-terminal part of LACRT (comprising the sequence part 112–137 aa), which is exclusively responsible for these beneficial effects and seems to be indispensable for tear film collapse prevention [64]. Thereby, it is assumed that natural LACRT is enzymatically cleaved into various proteoforms during specific environmental conditions (e.g., bacterial infection) eliciting its prosecretory and mitogenic biofunctions [64,65]. Nonetheless, the N-glycosylations site at position 119 is not occupied in the bioactive peptide compound Lacripep^TM^ but might play an important regulatory function in the proteolytic cleavage of full-length LACRT to control its complex molecular mode of action.

In addition, we also identified other proteomic markers, such as the polymeric immunoglobulin receptor (PIGR) or immunoglobulin heavy constant α-1 (IGHA1), as reliable glycoproteins (see Table 1) which are important components of the ocular defense system and form the basis for the natural protective barrier [66,67]. However, the main objective of the present study was to develop an efficient lectin-based affinity technology to first enrich native glycoproteins from human tear sample pools. After the successful development and optimization of the affinity method, the established affinity technology was used for tear glycopeptide enrichment and resulted in the reliable MS-based identification of 11 tear N-glycoproteins with 26 characteristic N-glycosylation sites in the present study. These findings are of particular interest because any imbalances in the composition of the natural tear film proteome, as well as their characteristic N-glycosylation levels, might lead to irreversible tear film instabilities promoting the formation of chronic ocular surface diseases such as DES. Because of this, it is of great importance to understand the precise role of N-glycosylations in maintaining the homeostasis and function of the ocular surface and to potentially use this knowledge for targeted medical intervention in the future. Furthermore, the intact glycopeptide characterization by MS [68] could also be a promising approach for ocular biomarker discovery in the future to obtain important structural and functional information about the attached N-glycan motifs.

## 4. Materials and Methods

### 4.1. Study Samples

For the development of the lectin-based affinity enrichment method, we first isolated the native glycoproteins from human tear film samples. For this part of the analysis, we used the tear samples of 2 healthy individuals (♂: 1 and ♀: 1, age: 30 ± 5 years). After the successful establishment of the method, we applied the optimized affinity technology to enrich glycopeptides from the human tear film samples. Therefore, the tear samples of 12 healthy volunteers (♂: 6 and ♀: 6, age: 31 ± 5 years) were included in this experiment. For individual tear sample collection, the Schirmer’s strip was placed onto the lower eyelid for 5 min; the flow distance of each strip was required to be >10 mm to exclude dry eye syndrome (DES). Subsequently, the tear fluid-containing strips were transferred into 1.5 mL reaction tubes and stored at −20 °C before further processing. All study participants did not show clinical signs of eye disease and contact lens wearers were excluded from the analysis. The investigation was approved by Landesärztekammer Rheinland-Pfalz (approval code ID: 837.027.17 (10861)) and was conducted in accordance with the tenets of the Declaration of Helsinki. In addition, informed consent was obtained from all study participants prior to tear sample collection.

### 4.2. Tear Sample Preparation and Pooling

For tear protein extraction, each reaction tube including the respective Schirmer’s strip was filled with 300 µL of phosphate-buffered saline (PBS) and treated at 4 °C overnight with gentle mixing. The next day, all samples were centrifuged at 1000× *g* for 1 min and the protein-containing supernatant was transferred into new reaction tubes. To determine the protein concentration of each sample, we used the Pierce™ BCA Protein Assay Kit (Thermo Fisher Scientific, Rockford, IL, USA) in combination with the Multiscan Ascent photometer (Thermo Fisher Scientific, Rockford, IL, USA) according to the manufacturer’s instructions. Afterward, the tear sample pools were prepared according to the previous specifications (see method Section 4.1) and a total protein amount of 100 µg per replicate was used for glycoprotein and/or glycopeptide enrichment. In the case of the glycoprotein enrichment for method development, we pooled the tear samples of two healthy individuals in equal shares to 100 µg aliquots and refilled each to a total volume of 200 µL with PBS (tear protein concentration: 0.5 µg/µL). For the glycopeptide enrichment, we pooled the individual tear samples of 12 healthy volunteers to create three biological replicates (100 µg of tear protein for each enrichment). To avoid gender-specific differences in the analysis, each biological replicate comprised the tear samples of two male and two female individuals in equal shares (25 µg of each sample) and was also refilled to a total volume of 200 µL with PBS. Since the detection of protein N-glycosylation sites by MS is based on enzymatic-induced modifications, we also included a master pool replicate (CTRL) without enzymatic treatment (PNGase F) in our analysis. Thereby, the CTRL replicate contained 100 µg of tear protein from all 12 healthy study participants in equal shares. The pooled tear samples ± pre-treatment were subjected to further glycoprotein/glycopeptide enrichment via lectin-based affinity techniques.

### 4.3. Lectin-Based Affinity Enrichment

For the glycoprotein/glycopeptide enrichment of the pooled tear film samples (see method Section 4.2), we used the agarose-bound lectins concanavalin (ConA), wheat germ agglutinin (WGA), jacalin (JAC), and ulex europaeus agglutinin I (UEA I) from the manufacturer Vector Laboratories (Burlingame, CA, USA). The glycan specificities of the respective lectins are listed in Table 2. For the enrichments, the agarose-bound lectins were either employed as single-lectin columns or as multi-lectin columns (comprising ConA/WGA/JAC termed 3L, or ConA/WGA/JAC/UEA I termed 4L) using 0.8 mL Pierce™ centrifuge column devices (Thermo Fisher Scientific, Rockford, IL, USA). These are empty, disposable microcentrifuge spin columns with a pore size of 30 µm to retain the agarose-bound lectins and are frequently used for gravity- or pressure-based chromatographic methods. For the enrichment of the native glycoproteins, the tear proteins were not pre-processed before affinity enrichment and the evaluation of the data was assessed by 1D gel electrophoresis and specific protein stainings (see method Section 4.4). Then, the successfully established affinity purification method was applied to the enrichment of glycopeptides followed by mass spectrometric (MS) analysis. Therefore, the tear sample pools were pre-treated by in-solution trypsin digest before affinity purification and subjected to further sample preparation protocols prior to MS analysis (see method Section 4.5).

Prior to affinity enrichment, the 0.8 mL Pierce™ centrifuge column devices were pre-filled with 200 µL of the respective agarose-bound lectin solution (ConA, WGA, JAC, or UEA I). Thereby, the columns were either filled with one lectin (200 µL) or with several lectins in equal shares to a total volume of 200 µL (3L- or 4L-column). Accordingly, the pre-filled column devices were inserted into 1.5 mL reaction tubes and centrifuged at 1000× *g* for 1 min. The flow-through (FT) fractions of the columns were discarded and the agarose-bound lectins were washed twice with 200 µL of PBS following the same procedure (centrifugation at 1000× *g* for 1 min). Then, the purified lectin columns were either filled with 200 µL of unprocessed tear sample pool (protein concentration: 0.5 µg/µL) or with 200 µL of tryptic tear sample digest (peptide concentration: 0.5 µg/µL, see method Section 4.5). The enrichment of the tear glycoproteins/glycopeptides was performed at 4 °C overnight with gentle mixing. The next day, the FT fractions were stored in new 1.5 mL reaction tubes and the remaining attached glycoproteins/glycopeptides were washed twice with 200 µL of PBS. Then, the respective glycoproteins/glycopeptides were eluted twice with 100 µL of 1% trifluoroacetic acid (TFA) for 10 min at RT with gentle mixing. Each eluate fraction was neutralized with 10 µL of 1M Tris HCl (pH 8.5) and afterward pooled in a new 1.5 mL reaction tube. Finally, the pooled eluate fractions (220 µL per sample) were evaporated in the speed vacuum concentrator (SpeedVac; Eppendorf, Darmstadt, Germany) at 45 °C until dryness before further processing.

### 4.4. Sample Preparation for the Enrichment of Tear Glycoproteins

#### 4.4.1. PNGase F Digest of Glycoproteins

In order to release the N-glycans from the glycoproteins, we digested some of the tear sample pools (100 µg; see method Section 4.2) with endoglycosidase PNGase F. This was performed to serve as an additional control sample during the affinity enrichment method development. Therefore, the tear sample pools were first evaporated in the SpeedVac at 45 °C until dryness and resolved in 20 µL of PBS using an ultrasonic bath for 10 min on ice. Afterward, 20 µL of 2% sodium dodecyl sulfate (SDS) solution was added to each sample and incubated for 30 min at 60 °C to denature the protein structure (reduce disulfide bonds). Subsequently, 20 µL of 4% nonidet P40 (NP-40) in 5X PBS was added to each sample followed by the addition of 4 µL of PNGase F solution (500 Units/mL; Roche Diagnostics, Mannheim, Germany). The digestion was performed at 37 °C overnight. The next day, all samples were evaporated in the SpeedVac at 45 °C until dryness before further processing.

#### 4.4.2. 1D Gel Electrophoresis

To evaluate the efficiency of the lectin-based affinity enrichment on the protein level, we performed 1D gel electrophoresis of the enriched tear glycoproteins (single-lectin and multi-lectin columns). Hence, the concentrated eluate fractions (see method Section 4.3) as well as the concentrated tear samples ± PNGase F (see method Section 4.4.1) were resolved in 5 µL of NuPage^TM^ LDS sample buffer (4X; Thermo Fisher Scientific, Rockford, IL, USA) and 2 μL of NuPage^TM^ reducing agent buffer (10X; Thermo Fisher Scientific, Rockford, IL, USA). Afterward, all samples were filled to a total volume of 20 µL with LC-MS grade water and incubated for 10 min at 70 °C to denature the proteins. In between, 10-well NuPAGE 12% or 4–12% Bis-Tris minigels (Thermo Fisher Scientific, Rockford, IL, USA) were assembled into XCell SureLock™ Mini-Cell Electrophoresis System (Invitrogen, Carlsbad, CA, USA) using the NuPAGE™ MES SDS Running Buffer 20X (Thermo Fisher Scientific, Rockford, IL, USA) according to the manufacturer’s protocol. All denatured protein samples were loaded onto the gels and 5 µL of the Seeblue^TM^ Plus 2 Pre-Stained Protein Standard (Thermo Fisher Scientific, Rockford, IL, USA) was added as a molecular weight marker. For the glycoprotein staining, we also included a glycosylated protein standard (horseradish peroxidase) as a positive control and a non-glycosylated protein standard (soybean trypsin inhibitor) as a negative control (see method Section 4.4.3). After sample loading, the protein mixtures as well as protein standards were separated for 1.5 h at 4 °C and 150 V.

#### 4.4.3. Protein Stainings (Glycoproteins and Coomassie)

For the detection of glycoproteins, we used the Pierce^TM^ Glycoprotein Staining Kit (Thermo Fisher Scientific, Rockford, IL, USA), and for the visualization of the total protein amount we used the Invitrogen™ Colloidal Blue Staining Kit (Thermo Fisher Scientific, Rockford, IL, USA). First, the proteins separated by 1D gel electrophoresis (see method Section 4.4.2) were fixed in 100 mL of 50% methanol (MeOH) for 10 min at RT. Subsequently, we performed either the glycoprotein staining and/or the Coomassie staining according to the supplier’s protocol. In general, the glycoprotein staining is based on the periodic acid-Schiff (PAS) reaction in which the glycols of the glycan structures are oxidized to reactive aldehyde groups using periodic acid as an oxidizing reagent. Then, the reactive aldehyde groups react with the Schiff’s reagent (fuchsin dye), which specifically stains the glycan structures of the glycoproteins on the gel and visualizes them as magenta bands with a light pink background. On the other hand, the Coomassie protein staining relies on the reversible molecular interaction of the dye (Coomassie) with the basic amino acid residues of the proteins and visualizes them as brilliant blue bands on the gel. However, after execution of the respective protein staining, the gels were destained for at least 16 h in deionized water (Coomassie-stained gels) or 3% acetic acid (glycoprotein-stained gels). Afterward, the respective stained gels were scanned with the Epson Perfection V600 Photo Scanner (Seiko Epson Corporation, Suma, Nagano, Japan) at a resolution of 700 dpi. For protein identification, the 1D gels were subjected to further in-gel trypsin digestion as described elsewhere [69,70] and measured by LC-MS (see method Section 4.5.4).

### 4.5. Sample Preparation for the Enrichment of Tear Glycopeptides

#### 4.5.1. In-Solution Trypsin Digestion

Prior to glycopeptide enrichment (see method Section 4.3), the tear sample pools containing 100 µg of protein were subjected to in-solution trypsin digestion (*n* = 3 biological replicates). First, the tear sample pools were evaporated in the SpeedVac at 45 °C until dryness and resolved in 100 µL of 10 mM ammonium bicarbonate (ABC) using an ultrasonic bath for 10 min on ice. Then, 30 µL of 20 mM dithiothreitol (DTT) in 10 mM ABC was added to each sample followed by incubation at 56 °C for 30 min. Afterward, 30 µL of 40 mM iodoacetamide (IAA) in 10 mM ABC was added to the sample pools and treated for 30 min at RT in the dark. As a final step, 50 µL of trypsin solution (Promega, Madison, WI, USA; 0.1 μg/μL in 10 mM ABC 10% acetonitrile (ACN)) were added to the sample pools and the reduced and alkylated proteins were digested overnight for at least 16 h at 37 °C. The next day, all samples were evaporated in the SpeedVac at 45 °C until dryness.

#### 4.5.2. Peptide Purification and Concentration Determination

Peptide purification via the SOLAµ™ SPE HRP plates (Thermo Fisher Scientific, Rockford, IL, USA) was performed after the in-solution trypsin digest (see method Section 4.5.1) as well as prior to the MS analysis (see method Section 4.5.3). First, the crude peptide digest was dissolved in 100 µL of 0.1% formic acid (FA) and treated for 10 min using an ultrasonic bath on ice. Activation of the SOLAµ^TM^ SPE membranes was performed with 100 µL of ACN followed by equilibration with 100 µL of 0.1% FA. After each step, the SOLAµ^TM^ SPE spin plate was centrifuged at 2,000 *g* for 1 min and the FT/eluate fractions were collected in 96-well microtiter microplates (Costar Corning Incorporated, Corning, NY, USA). The SOLAµ^TM^ SPE membranes were loaded twice with the crude sample digest followed by five washing steps with 100 µL of 0.1% FA. Finally, the concentrated and purified peptides were eluted twice with 100 µL of 0.1% FA in 50% ACN and both eluate fractions were pooled in a new reaction tube (200 µL per sample). Finally, the purified peptide samples were evaporated in the SpeedVac at 45 °C to dryness before further processing. In addition, the peptide concentration of each sample was determined with the Pierce^TM^ Quantitative Colorimetric Peptide Assay Kit (Thermo Fisher Scientific, Rockford, IL, USA) in combination with the Multiscan Ascent photometer according to the supplier’s instructions. Specifically, this was performed after glycopeptide affinity enrichment from the purified tear peptide digests (see method Section 4.3).

#### 4.5.3. PNGase F Digest of Glycopeptides

To release the N-glycans from the glycopeptides, we treated the enriched tear glycopeptides (see method Section 4.3) with endoglycosidase PNGase F, enabling the detection of their specific N-glycosylation sites by MS. The enriched and concentrated tear glycopeptides were dissolved in 200 µL of 5× PBS and subsequently treated for 10 min using an ultrasonic bath on ice. Then, 1.5 µL of the PNGase F solution (500 Units/mL) was added to each sample and digested at 37 °C overnight. The next day, the deglycosylated peptides were evaporated in the SpeedVac at 45 °C until dryness. Prior to MS analysis, the modified (deamidated) glycopeptides were concentrated and purified with the SOLAµ™ SPE HRP plates as described in detail in method Section 4.5.2.

#### 4.5.4. MS Analysis

For the MS measurements of the tear glycopeptides ± PNGase F, we used the hybrid linear ion trap-Orbitrap MS system (LTQ Orbitrap XL, Thermo Fisher Scientific, Rockford, IL, USA), which was online coupled to the EASYnLC 1200 system (Thermo Fisher Scientific, Rockford, IL, USA). The enriched and purified glycopeptides ± PNGase F were dissolved in 0.1% FA to a peptide concentration of 0.5 µg/µL, and 2 µL of this mixture was injected into the system for each run. The chromatographic separation of the peptides was performed with the PepMap C18 column system (75 µm × 150 mm; Thermo Fisher Scientific, Rockford, IL, USA) and the flow rate of the nanoLC system was set to 0.3 µL/min. The running buffer A was comprised of 0.1% FA and the running buffer B consisted of 0.1% FA in 80% ACN. The peptides were eluted within 90 min using the following solvent gradient: 5–30% B (0–60 min), 30–100% B (60–80 min), and 100% B (80–90 min). Each sample was measured seven times in total with different MS settings (summarized in Table 3) in order to maximize the identification rates of all potential N-glycosylation sites. All MS data have been deposited to the ProteomeXchange Consortium via the PRIDE partner repository [71] with the dataset identifier PXD038280.

#### 4.5.5. Data Analysis and Bioinformatics

For the identification of the tear glycoproteins and their respective N-glycosylation sites, we used the bioinformatics software PEAKS Studio X (Bioinformatics Solutions Inc., Waterloo, ON, Canada). This software platform encompasses a state-of-the-art de novo sequencing-assisted database search, which significantly increases the sequence coverage of proteins, and also features a state-of-the-art PTM search algorithm to maximize N-glycosylation site identifications. The database-assisted protein identification was performed with the SwissProt database and the taxonomy *Homo sapiens* (date: 13 April 2021, number of sequences: 20,408) using the following search parameters: peptide ion mass tolerance of ±30 ppm, fragment ion mass tolerance of 0.5 Da, tryptic cleavage, maximum of two missed cleavage sites, carbamidomethylation as fixed modification, and acetylation (N-terminal protein), oxidation, and deamidation of asparagine (N) as variable modifications. Due to the release of the N-glycans from the N-glycopeptides by PNGase F, the enzymatic-induced modification (deamidation) marked the original glycosylation site in the peptide/protein and was detectable by MS. All protein/peptide identifications were filtered with a false discovery rate (FDR) < 1%. To identify only true N-glycosylation sites, we only considered deamidation sites, which were detected with an Ascore ≥ 20 in each peptide. The Ascore is a software-specific (PEAKS) localization score assigned to modifications on the peptide. It is a -log10-adjusted *p*-value and describes the probability of a PTM site occurring at the reported position compared to other possible positions. A -log10(*p*)-adjusted value of 20 is equal to a *p*-value of 0.01. In addition, each deamidation site had to exhibit the canonical protein sequence motif N–X–serine (S)/threonine (T)/cysteine (C), where X indicates any aa except proline (P). Since deamidation sites can also occur as natural PTMs or might be chemically induced during sample preparation, we also included enriched, non-modified N-glycopeptides (CTRL group) in this analysis. The respective deamidation site was only acknowledged as a reliable and true protein N-glycosylation site (enzymatic-induced by PNGase F) if it was not detected in the untreated CTRL group.

## Figures and Tables

**Figure 1 molecules-28-00648-f001:**
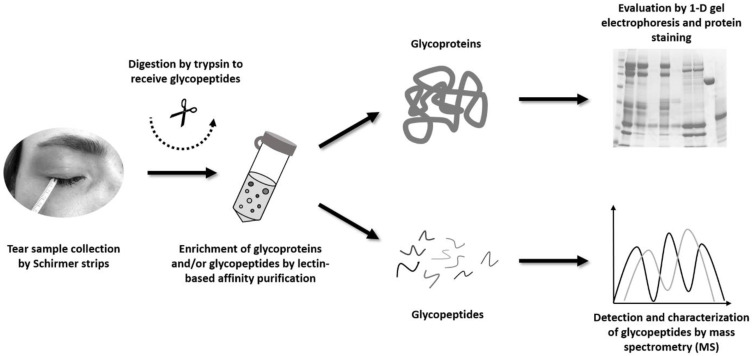
Schematic illustration of the sample preparation protocol used for the affinity enrichment of glycoproteins/glycopeptides from human tear sample pools. The tear samples from healthy individuals were collected by Schirmer’s strips and the tear proteins were subsequently eluted with 300 µL of PBS. Then, the individual tear protein samples were pooled according to the specifications in the method Section 4.3. The enrichment of the glycoproteins and/or glycopeptides was based on lectin affinity chromatography. The enrichment performance of various single-lectin and multi-lectin columns was evaluated first on the native glycoprotein level by 1D gel electrophoresis and specific protein stainings (glycoprotein and Coomassie staining). Next, the established and optimized lectin affinity method was employed for the enrichment of glycopeptides from human tear sample digests. Then, the enriched and concentrated glycopeptides and their characteristic N-glycosylation sites were identified by high-resolution MS.

**Figure 2 molecules-28-00648-f002:**
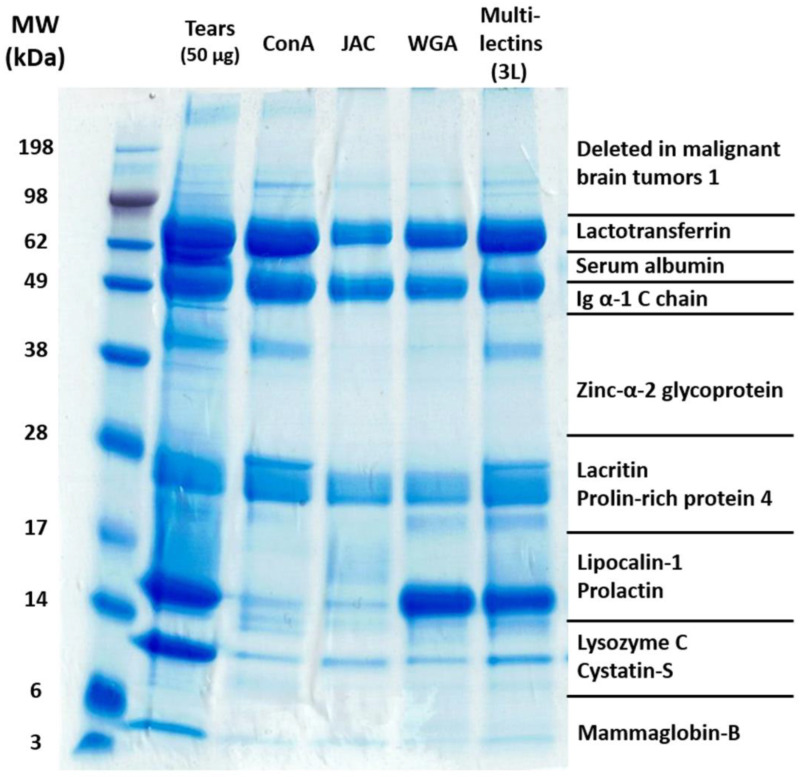
The 1D gel electrophoresis and Coomassie protein staining of potentially enriched glycoproteins from human tear sample pools. The image shows the eluate fractions of single-lectin columns (ConA, JAC, and WGA) and the multi-lectin column device (combination of ConA, JAC, and WGA, termed 3L). An untreated tear sample pool (50 µg) was included in this experiment to match potentially enriched glycoproteins with the native tear film proteome. The most abundant protein markers were labeled in the respective mass range.

**Figure 3 molecules-28-00648-f003:**
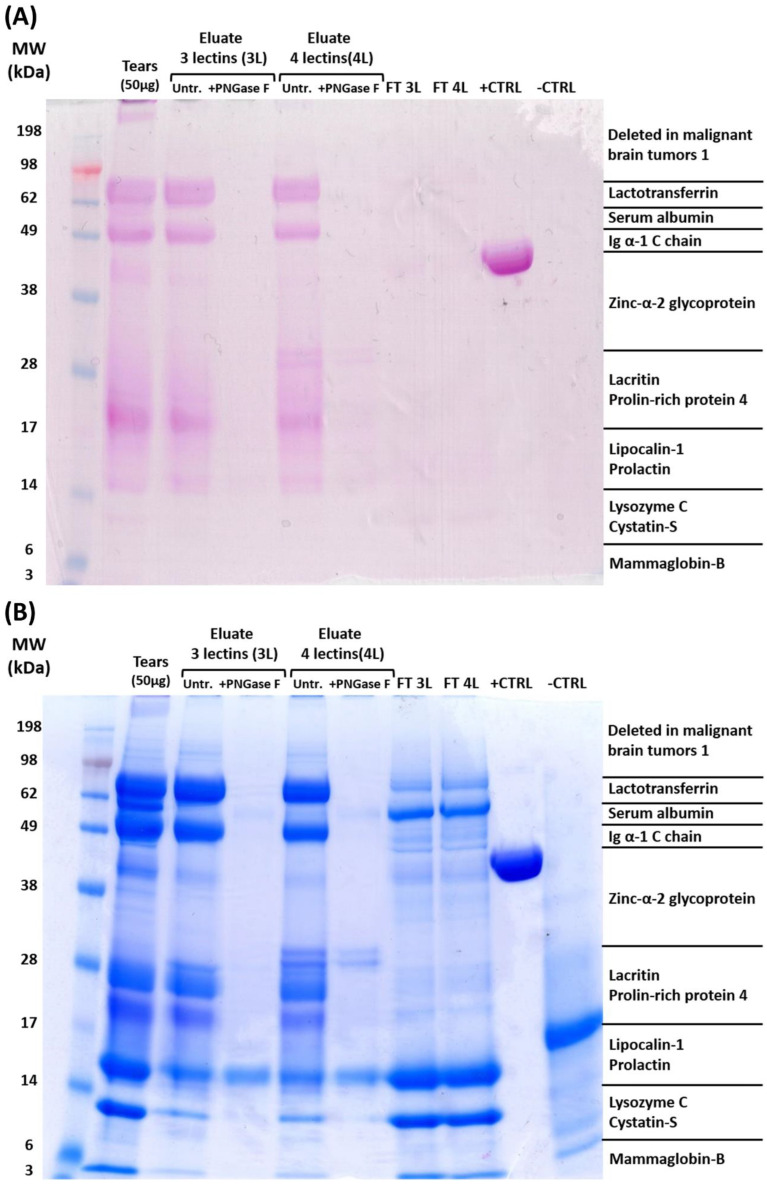
The 1D gel electrophoresis and glycoprotein/Coomassie staining of enriched glycoproteins from human tear sample pools. Both images illustrate the eluate as well as the flow-through (FT) fractions of the 3L multi-lectin column (combination of ConA, JAC, and WGA) and the 4L multi-lectin column system (combination of ConA, JAC, WGA, and UEA I). Prior to affinity enrichment, the tear sample pools were partially pre-treated with PNGase F to release the N-glycans from the glycoproteins. An untreated tear sample pool (50 µg) was also included to match the enriched glycoprotein spots with the native tear film proteome. In addition, a glycosylated protein standard (horseradish peroxidase, +CTRL), as well as a non-glycosylated protein standard (soybean trypsin inhibitor, -CTRL), were either included as a positive or negative control in this experiment. (**A**) Glycoprotein staining of the 1D gel which specifically colored the glycan motifs of the tear film glycoproteins/protein standards as magenta stained protein bands. (**B**) Coomassie protein staining of the 1D gel to visualize all the proteins as brilliant blue protein bands. The most abundant protein markers were labeled in the respective mass range.

**Figure 4 molecules-28-00648-f004:**
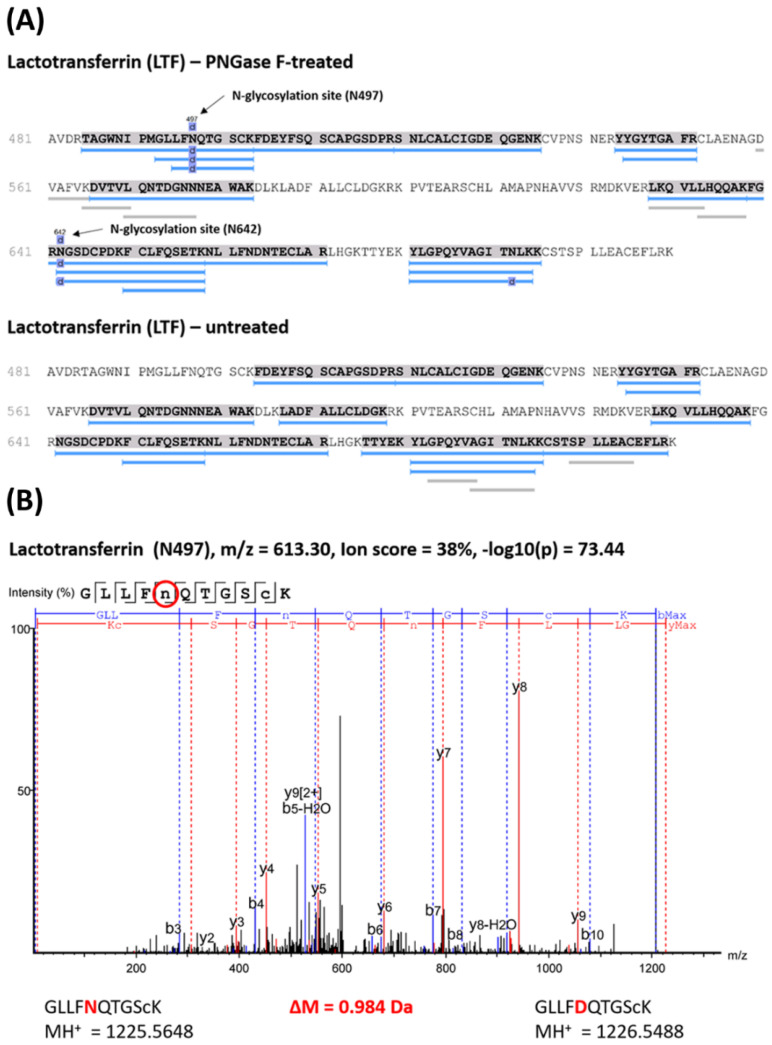
The MS-based detection of tear glycoprotein lactotransferrin (LTF) and characterization of its specific N-glycosylation sites. (**A**) The identified sequence of enriched glycoprotein LTF which was either pre-treated with PNGase F or served as an untreated control (CTRL) group. The enzymatic-induced N-glycosylation sites N497 and N642 are marked in the respective sequence parts of LTF + PNGase F treatment but are not present in the untreated CTRL group. (**B**) The MS/MS spectrum of an N-glycopeptide from LTF indicating the glycosylation site at N497. The red-colored amino acid within the sequence marks the N-glycosylation site which was deamidated from asparagine (N) to aspartic acid (D) due to the enzymatic release of the N-glycan. This enzymatic-induced modification results in a mass shift of +0.984 Da which can be detected by high-resolution MS.

**Table 1 molecules-28-00648-t001:** Glycoproteins and their N-glycosylation sites in the human tear sample pools of healthy individuals.

Protein ID	Protein Name	Gene Name	N-glycosylation Sites	Present in R1	Present in R2	Present in R3	CTRL Group	Reported in Literature	Peptide Score
P02788	Lactotransferrin	LTF	N497	√	√	√	-	[24,25,26,27]	103
			N642	√	√	√	-		87
P01833	Polymeric immunoglobulin receptor	PIGR	N83	√	√	√	-	[24,25,28,29]	106
			N90	√	√	√	-		105
			N135	√	√	√	-		73
			N186	√	√	√	-		96
			N421	√	√	-	-		112
			N469	√	√	√	-		108
			N499	√	√	-	-		108
P01877	Immunoglobulin heavy constant alpha 2	IGHA2	N47	√	√	√	-	[24,26,30]	54
			N92	-	√	-	-		102
			N131	√	√	√	-		108
			N205	√	√	√	-		81
P01876	Immunoglobulin heavy constant alpha 1	IGHA1	N144	√	√	√	-	[25,29,31,32]	108
			N340	√	√	√	-		101
P25311	Zinc-α2-glycoprotein	AZGP1	N109	√	√	√	-	[24,25,26]	120
			N112	√	√	-	-		77
			N128	√	√	√	-		70
Q9GZZ8	Extracellular glycoprotein lacritin	LACRT	N119	√	√	√	√	[25,28]	88
P01591	Immunoglobulin J chain	JCHAIN	N67	√	√	√	-	[24,25,28,29]	93
P12273	Prolactin-inducible protein	PIP	N105	√	√	√	-	[25,28,33,34]	92
P02787	Serotransferrin	TF	N630	n.d.	√	√	-	[25,26,28,29]	90
P00738	Haptoglobin	HP	N207	n.d.	√	n.d.	-	[25,28,29]	85
			N211	n.d.	√	n.d.	-		85
			N241	n.d.	√	n.d.	-		103
P01009	Alpha-1-Antitrypsin	SERPINA1	N271	n.d.	√	n.d.	-	[25,26,29,30]	94

√: Present and significant (Ascore ≥ 20), -: N-glycosylation site not present, n.d.: Protein was not detectable.

**Table 2 molecules-28-00648-t002:** List of lectins and their respective glycan specificity.

Lectin	Glycan Specificity
Concanavalin (ConA)	α-linked mannose
Wheat germ agglutinin (WGA)	N-acetylglucosamine
Jacalin (JAC)	galactosyl (ß-1,3) N-acetylgalactosamine
Ulex europaeus agglutinin I (UEA I)	α-linked fucose

**Table 3 molecules-28-00648-t003:** List of MS parameters used for the identification of N-glycosylation sites.

No.	Fragmentation Method	Dynamic Exclusion Enabled	Repeat Count	Repeat Duration (s)	Exclusion Size List	Exclusion Duration (s)	Automatic Gain Control (AGC)
1	CID	yes	1	30	100	180	1 × 10^6^
2	CID	yes	1	30	100	180	5 × 10^5^
3	CID	yes	1	30	50	180	1 × 10^6^
4	CID	yes	1	30	100	180	1 × 10^6^
5	CID	yes	1	30	100	300	1 × 10^6^
6	CID	yes	1	30	100	90	1 × 10^6^
7	HCD	no	-	-	-	-	1 × 10^6^

**CID:** Collision-induced dissociation **HCD:** Higher energy collision-induced dissociation.

## Data Availability

All MS data have been deposited to the ProteomeXchange Consortium via the PRIDE partner repository with the dataset identifier PXD038280.

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
