# Peer review of "Lectin-Based Affinity Enrichment and Characterization of N-Glycoproteins from Human Tear Film by Mass Spectrometry"

_molecules, 2023, doi:10.3390/molecules28020648_

Round 1
Reviewer 1 Report
In the present study, the authors aimed to develop a lectin-based affinity method for the enrichment and concentration of tear glycoproteins/glycopeptides and the characterization of their specific N-glycosylation sites by high-resolution mass spectrometry (MS). For method development and evaluation, they accumulated native glycoproteins from human tear sample pools and assessed the enrichment efficiency of different lectin column systems by 1D gel electrophoresis and specific protein stainings (Coomassie and glycoproteins). The best-performing multi-lectin column system (comprising the four lectins ConA, JAC, WGA and UEA I, termed as 4L) was applied for glycopeptide enrichment from human tear sample digests followed by MS-based detection and localization of specific N-glycosylation sites. Their study identified in total 26 N glycosylation sites of 11 N-glycoproteins in tear sample pools of healthy individuals.
Specific points:
1) The application is interesting, although the technology is not novel. As we know enrichment with lectin(s) has been a traditional way for glycoprotein/glycopeptide enrichment. It would be great such classic papers can be cited (e.g., https://pubmed.ncbi.nlm.nih.gov/12754521/).
2) Although PNGase F-based N-glycopeptide site mapping is still used sometimes, it suffers from shortcomings (e.g., potential false positives). Analysis of intact glycopeptides has been made possible in recent years, which can provide much enriched information of the glycoproteins/glycopeptides/glycosites if used. It would be great if the authors can at least comment on these aspects in their article as (e.g., as one future direction).
Author Response
In the present study, the authors aimed to develop a lectin-based affinity method for the enrichment and concentration of tear glycoproteins/glycopeptides and the characterization of their specific N-glycosylation sites by high-resolution mass spectrometry (MS). For method development and evaluation, they accumulated native glycoproteins from human tear sample pools and assessed the enrichment efficiency of different lectin column systems by 1D gel electrophoresis and specific protein stainings (Coomassie and glycoproteins). The best-performing multi-lectin column system (comprising the four lectins ConA, JAC, WGA and UEA I, termed as 4L) was applied for glycopeptide enrichment from human tear sample digests followed by MS-based detection and localization of specific N-glycosylation sites. Their study identified in total 26 N glycosylation sites of 11 N-glycoproteins in tear sample pools of healthy individuals.
Specific points:
1) The application is interesting, although the technology is not novel. As we know enrichment with lectin(s) has been a traditional way for glycoprotein/glycopeptide enrichment. It would be great such classic papers can be cited (e.g., https://pubmed.ncbi.nlm.nih.gov/12754521/).
Response: The respective reference was added in the discussion section in line 282.
2) Although PNGase F-based N-glycopeptide site mapping is still used sometimes, it suffers from shortcomings (e.g., potential false positives). Analysis of intact glycopeptides has been made possible in recent years, which can provide much enriched information of the glycoproteins/glycopeptides/glycosites if used. It would be great if the authors can at least comment on these aspects in their article as (e.g., as one future direction).
Response: We agree with the reviewer that nowadays the analysis of intact glycopeptides is feasible. However, these methodological approaches require faster MS instrumentations, multiple fragmentation methods (e.g., ETD) and/or advanced bioinformatics solutions. Nevertheless, we shortly commented these aspects in the discussion section in line 438-440.
Reviewer 2 Report
Overall, this paper is well-written and presents a novel method that can be applied to an emerging biological fluid for non-invasive disease screening. The results and conclusions are presented clearly for the most part.
I would suggest some minor revisions before publication as follows:
Figure 2 is a bit confusing. It looks as if the labeled protein bands were stronger in the raw tears compared to the column enriched fractions. Can the authors explain this? In addition, it is unclear as to what "Deleted in malignant brain tumors 1" refers to. Are you referring to as yet unidentified protein biomarkers?
The stated goal of the paper is the focus on the affinity chromatography method. It is unclear to me whether commercial columns were used or custom-made. The methods/materials section that describes the process (4.3) could benefit from a schematic diagram to help the reader visualize the novel centrifugal columns. The way the process is described I am not sure whether the centrifugal process was applied to the protein purification or just for the column preparation. Because this is the authors stated main purpose, I think further descriptions along with helpful schematics or figures are warranted.
Author Response
Overall, this paper is well-written and presents a novel method that can be applied to an emerging biological fluid for non-invasive disease screening. The results and conclusions are presented clearly for the most part.
I would suggest some minor revisions before publication as follows:
Figure 2 is a bit confusing. It looks as if the labeled protein bands were stronger in the raw tears compared to the column enriched fractions. Can the authors explain this? In addition, it is unclear as to what "Deleted in malignant brain tumors 1" refers to. Are you referring to as yet unidentified protein biomarkers?
Response: We agree with the reviewer that some of the labeled proteins were stronger in the raw tears compared to the enriched glycoproteins (e.g., Lactotransferrin). One possible explanation for this could be that not all proteoforms of the respective protein are glycosylated and are accordingly not enriched by lectin-based chromatography. No, the protein „Deleted in malignant brain tumors 1“ is not an unidentified protein marker, but was identified with high confidence as abundant tear biomarker in the marked mass range in previous studies of our group (Perumal et al., 2014 and 2015).
The stated goal of the paper is the focus on the affinity chromatography method. It is unclear to me whether commercial columns were used or custom-made. The methods/materials section that describes the process (4.3) could benefit from a schematic diagram to help the reader visualize the novel centrifugal columns. The way the process is described I am not sure whether the centrifugal process was applied to the protein purification or just for the column preparation. Because this is the authors stated main purpose, I think further descriptions along with helpful schematics or figures are warranted.
Response: We used the 0.8 mL Pierce™ centrifuge column devices (purchased from Thermo Fisher) and filled them in equal shares with the agarose-bound lectins ConA, WGA, JAC and UEA I (purchased from Vector Laboratories) as single-lectin or multi-lectin columns. The glycoprotein/glycopeptide enrichment is an centrifugal process. We revised the method section 4.3 according to the recommendations of the reviewer.